# Predicting foot orthosis deformation based on its contour kinematics during walking

**Maryam Hajizadeh**[1]*, **Benjamin Michaud**[2], **Gauthier Desmyttere**[2], **Jean-Philippe Carmona**[3], **Mickaël Begon**[1,2]

**1** Laboratory of Simulation and Movement Modelling, Institute of Biomedical Engineering, Faculty of Medicine, Université de Montréal, Montréal, Quebec, Canada, **2** Laboratory of Simulation and Movement Modelling, School of Kinesiology and Physical Activity Sciences, Faculty of Medicine, Université de Montréal, Montréal, Quebec, Canada, **3** Caboma, Montréal, Quebec, Canada

* maryam.hajizadeh@umontreal.ca

**Data Availability Statement:** All relevant data are within the paper and its Supporting Information files.

## Abstract

### Background

Customized foot orthoses (FOs) are designed based on foot posture and function, while the interaction between these metrics and FO deformation remains unknown due to technical problems. Our aim was to predict FO deformation under dynamic loading using an artificial intelligence (AI) approach, and to report the deformation of two FOs of different stiffness during walking.

### Methods

Each FO was fixed on a plate, and six triad reflective markers were fitted on its contour, and 55 markers on its plantar surface. Manual loadings with known magnitude and application point were applied to deform "sport" and "regular" (stiffer) FOs in all regions (training session). Then, 13 healthy male subjects walked with the same FOs inside shoes, where the triad markers were visible by means of shoe holes (walking session). The marker trajectories were recorded using optoelectronic system. A neural network was trained to find the dependency between the orientation of triads on FO contour and the position of markers on its plantar surface. After tuning hyperparameters and evaluating the performance of the model, marker positions on FOs surfaces were predicted during walking for each subject. Statistical parametric mapping was used to compare the pattern of deformation between two FOs.

### Results

Overall, the model showed an average error of <0.6 mm for predicting the marker positions on both FOs. The training setup was appropriate to simulate the range of triads' displacement and the peak loading on FOs during walking. Sport FO showed different pattern and significantly higher range of deformation during walking compared to regular FO.

**Funding:** This manuscript is a part of research project which is financially supported by NSERC (Natural Sciences and Engineering Research Council) R&D Coop with industrial partners Medicus and Caboma, and MEDTEQ under grant number RDCPJ 506194-16, principal researcher: Mickaël Begon (MB), original project title: "FOOT¡ (Functional Optimized Orthotic Trabecular Insole): Une orthèse plantaire personnalisée selon la dynamique du pied pour l'impresson 3D" ("customized foot orthosis to meet the demands of foot dynamics for 3D printing"), and URL funder: https://www.nserc-crsng.gc.ca/index_eng.asp. In addition, FRQNT (The Fonds de recherche du Québec - Nature et technologies) provided financial support in the form of doctoral research scholarships program for foreign students (DE), File No. 208405, awarded to Maryam Hajizadeh (MH), and URL funder: http://www.frqnt.gouv.qc.ca/en/accueil The funders, either NSERC, MEDTEQ or FRQNT, as well as our industrial partner Medicus did not have any role in the study design, data collection and analysis, decision to publish, or preparation of this manuscript. However, Jean-Philippe Carmona (JPC), a mechanical engineer in Caboma, was involved during our regular meetings and brought technical ideas to improve the algorithms and interpretation of results. JPC also contributed to reviewing and editing the paper.

**Competing interests:** It should also be mentioned that our innovative approach for quantifying the deformation of foot orthosis during walking does not provide any commercial asset at this point. Therefore, the materials presented in this manuscript is not related to any kind of consultancy, patents, products in development, or marketed products. Furthermore, our commercial affiliation does not alter our adherence to PLOS ONE policies on sharing data and materials. Finally, we confirm that there is no competing of interest, but a fruitful collaboration between our research team and our industrial partners, Caboma and Medicus. On-going studies involving flat feet patients may provide commercial advantages.

## Conclusion

Our technique enables an indirect and accurate estimation of FO surface deformation during walking. The AI model was capable to make a distinction between two FOs with different stiffness and between subjects. This innovative approach can help to optimally customize the FO design.

## Introduction

The human foot plays an important role in propulsion, stability and efficiency [1–3]. If the foot architecture cannot support the biomechanical demands of different activities, various foot pathologies might occur [4–7]. Foot orthoses (FOs) are getting more popular in clinics to treat several types of symptomatic feet [8]. FO comes into direct contact with the foot and is, therefore, subject to deformation during dynamic loading, such as walking. FO design and foot structure work in parallel to conduct the range and pattern of deformation. Therefore, the motion and function of symptomatic foot could be enhanced by managing FO deformation via FO design.

Previous literature suggests the dependency between FO design and different alterations in foot posture and pressure [9, 10]. FO with medial posting brings about lower ankle eversion during walking, while lateral posting exhibits a reverse effect [10, 11]. In addition, arch-conforming shape of FO as well as insole stiffness have been reported to exhibit important impact on reducing peak plantar pressure [12]. Heel lifts with higher thickness and material hardness lead to higher plantar pressure in forefoot and heel regions compared to medium and soft materials. Both inadequate support induced from softer heel lifts and decreased compliance from harder heel lifts could subsequently compromise dynamic stability and comfort [13]. Therefore, both shape and stiffness of FO could modify the altered foot motion pattern and plantar pressure related to pathologies.

While the interaction between FO design and foot motion and function has been already reported [14–16], the behavior of FO, *i.e.* its deformation, during dynamic loading remains unknown. The main issue is that it is not possible to directly capture FO deformation via optoelectronic system and reflective markers, since FO plantar surface is hidden by foot contact. Artificial intelligence (AI) and finite element analysis have the potential of estimating FO deformation as alternative techniques. AI has been increasingly implemented to accurately predict time series and sequences with complex patterns [17–19]. Since AI is a data-driven self-adaptive method, it needs much fewer humanly decided assumptions and simplifications than finite element analysis [17]. Through a training dataset, the AI responds to the information flowing into the network. A test session is then used to generate the output features as a response to previously unseen inputs, in order to assess the performance of generated model [19, 20]. Thereafter, the validated model would be used to predict the FO deformation during walking, where the FO plantar surface is hidden by foot contact. The predicted FO deformation with AI could finally be used to estimate objective function for a finite element model to optimize the design of FO and improve foot posture and plantar pressure of symptomatic foot.

The objective of this study was to suggest a novel method to predict the deformation on plantar surface of FO based on the orientation of FO contour during dynamic loading. To this end, a setup was designed to provide a comprehensive dataset for training an AI model. The dataset obtained from this setup could simulate the walking condition by controlling the orientation and magnitude of applied loads. The validation of the AI model was examined via the

test dataset. Finally, this model was used to predict the FO deformation during walking with two FOs of different stiffness. It was hypothesized that the AI model enabled us to differentiate between FOs in terms of different ranges or patterns of deformation.

## Materials and methods

A total of 13 healthy male subjects with normal feet (age = 25.9±4.2 years old, height = 176.2 ±4.3 cm, weight = 74.6±7.8 kg, shoe size 9–10) were recruited via call for volunteers at the School of Kinesiology. The inclusion criteria for participants was to be free from any limb injuries at the time of testing and having no known history of foot structural abnormalities or pathologies. The subjects were asked whether they have ever used foot orthosis or therapeutic insoles for any reason of pain or foot injuries especially flatfoot deformity. In addition, two observers, GD and MH, had to examine and include the subjects with normal medial arch during weight bearing/non weight bearing position and normal rearfoot orientation relative to tibia long axis. Ethical approval was obtained from University of Montreal (17-145-CERES-D approval), and all participants gave their written informed consent.

### Setup design and data acquisition

A three-dimensional scan of a positive cast mold generated from the average foot shape of 2000 European males (foot size 10) was used to design a customized three-quarter length FO. The FO plantar surface followed the contour shape of foot (medial and lateral arch, heel cup) with 1.5 mm thickness superimposed to honeycombs. The height of the honeycomb cells was then changed to reach two different stiffness (termed as "sport" *versus* "regular" FO). The regular FO was stiffer, *i.e.* less deformable, as an effect of higher honeycombs compared to sport FO. The design also included six double-cross slots on the FO contour allowing for fitting six triads, each consisting of three reflective markers mounted on branches of 20 mm in length. Both FOs and triads were 3D printed in nylon 12 (Fig 1a). Data collection was performed in two measurement sessions: training and walking.

In the training session, each FO was placed and fixed at the heel region on a wooden plate covered with a soft material corresponding to a shoe midsole property. Fifty-five 3-mm hemispherical retroreflective markers were taped to the plantar surface of FO, and triads were fitted on its contour (Fig 1a and 1b). After capturing an unloaded static position for each FO, a 20-cm long stick with a narrow circular shape at the tip (6-mm diameter) and equipped with a load cell (Model XLU68F-250, Full Scale Range 250 Lbs, Delta Metrics Inc., Worthington, Ohio) was used in order to deform FO. The load cell was primarily calibrated with compressive loads before training session (S1 Fig). Different loadings were applied to all FO regions using the stick. The application point and magnitude of applied load was controlled via four retroreflective markers placed on the stick and the load cell, respectively (Fig 1a).

During the walking session, each FO was placed inside standard sports shoes (New Balance 860 v8), and Medilogic WLAN insole was placed on plantar surface of FO to record foot plantar pressure. Six circular holes (25-mm diameter) were made on the upper shoe allowing a direct access to the FO contour in order to fit the triads. Each participant was asked to walk on a treadmill for 5 minutes for habituation, where his comfortable speed was acquired for the following measurements (S1 Table). Then, the participant walked for 3 minutes at this acquired speed for each sport and regular FO condition (Fig 1c). A rest period of 5 minutes was given between conditions to avoid fatigue effects. Data acquisition included the recording of walking on a treadmill for 3 minutes at self-selected comfortable speed for each subject with each FO (Fig 1c). The last 30-s of walking were used for further processing.

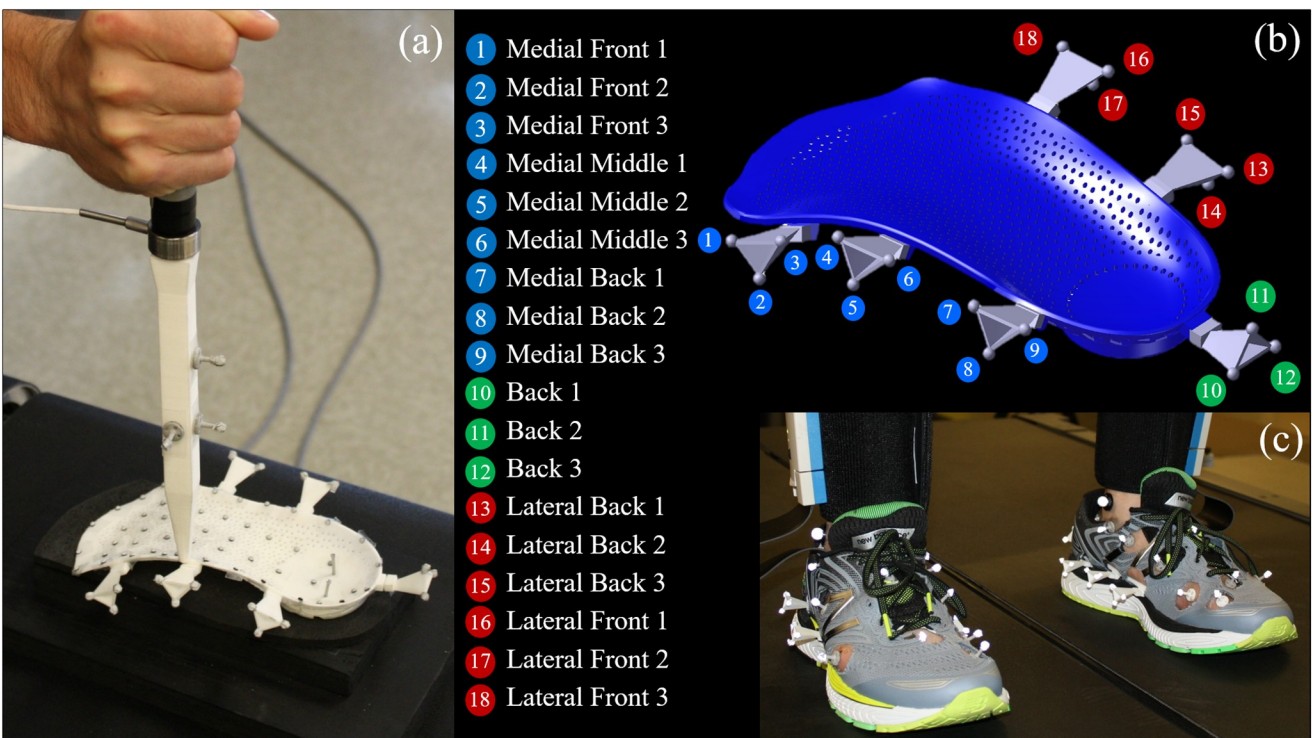

**Fig 1. Set-up and markerset for two sessions. (a)** *Training* session with attaching markers on plantar surface and triads on the contour of foot orthosis (FO), fixing FO on heel part, and load application. **(b)** The position and tag of triad markers fitted on foot orthosis contour. **(c)** *Walking* session with placing FO inside the shoe and inserting triads.

In both sessions, the markers' trajectories were recorded using an 18-camera VICON™ optoelectronic motion analysis system (Oxford Metrics Ltd, Oxford, UK) at a sampling rate of 100 Hz. In addition, the load cell data and foot plantar pressure were recorded at sampling rate of 1000 Hz and 400 Hz, respectively.

## Deep learning for predicting FO deformation

The standardization of data was performed by expressing the data in a local coordinate system at each time frame. This local coordinate frame was defined from the position of three markers on triads (markers 4, 11, and 18 in Fig 1b). The input data were the three-dimensional orientation of each triad relative to back triad, being computed by YXZ Cardan sequences. They were expressed as a vector of size OT = 15 (3 [relative orientation angle per triad] × 5 [triads]), trained over the number of time frames. The output feature was the position of markers on plantar surface of FO with the size of MPS = 165 (3 [$x, y, z$] × 55 Markers on Plantar Surface) over the number of time frames.

A large amount of data was acquired for each FO in order to let load application in several FO regions. The frames without loading ($< 0.1$ N) was omitted from the acquired data, and the remained frames were then stratified based on 10 different regions of load application and got shuffled. These regions were defined as medial front, medial middle, medial back, back medial, back, back lateral, lateral back, lateral middle, lateral front, and middle FO regions (S2

Fig). This method enabled us to control the proportion of different loadings over time and distribute them in the whole dataset.

The stratified shuffled data were split into learning set (85%) and test set (15%) for each FO. Data of *learning* set was used to tune the hyperparameters of the model, and *test* set was kept aside as unseen data to evaluate the performance of final selected model. Grid search algorithm with K-fold cross validation (K = 5) was used to tune the model. This algorithm performs exhaustive search to find the optimal parameters, where the 5-fold cross-validation splits the learning set into five different groups of train and validation sets to avoid overfitting. The tested hyperparameters were the number of layers at a range from 1 to 5, three optimizers namely "Adam", "Adadelta", and "Adagrad", learning rates of 0.01 and 0.005, batch size in the range of 16, 32, 64, and 128, and epoch in the range of 20, 50, and 100. The architecture of neural network was finally selected as densely connected neural network, with four layers (two hidden layers), optimizer "Adam" with learning rate of 0.005, loss function "mean square error", 100 epochs and batch size of 16, designed by TensorFlow [21]. These parameters provided the best accuracy (lowest loss) for learning the dependency between input (triads) and output (plantar surface markers).

**Validation.** A primary step for validation was to check whether the deformation and loading during the training session replicates what happens during walking. Regarding deformation, the ranges of displacement for 18 markers mounted on triads expressed in the local frame during *walking session* for all included subjects was compared to the corresponding ranges during *training* session for each sport and regular FO separately. This comparison was done for upward/downward displacement of triads, since it is the dominant component of displacement which can be considered for biomechanical behavior of FO. For loading, the range of peak plantar pressure for all included subjects during walking was compared to the range of manual loading from stick during training session for each region of FO. The more overlap exists between the ranges of displacement and loading during *training* and *walking* sessions, the more accurate predictions would be expected [22, 23]

In addition, the generalization performance of the neural network was evaluated with the *test* set for each FO. The root mean square error (RMSE) was calculated as the difference between the "measured" and "predicted" marker positions on the FO plantar surface; normalized RMSE (NRMSE) were also estimated by normalizing the RMSE to the maximal deformation for each marker, calculated as the Euclidian distance between the position of MPSs on test set and reference position, *i.e.* unloaded static position.

**Walking.** The position of markers on plantar surface of both sport and regular FO were predicted using relative orientation of triads during the walking session for each subject. Similar to the training session, the coordinates of all available markers were expressed in the coordinate system generated by the three selected markers on triads for both sessions. This projection could reflect both rigid transformation (rotation and translation) and deformation of FO in the shoes during walking. Therefore, polar decomposition was used to calculate the optimal roto-translation between each time frame and the static condition to only extract the deformation [24, 25]. Finally, the 3D positions of optimally transformed markers were normalized to stance phase of walking, and the depression/reformation of each marker was calculated by subtracting its upward/downward position from its corresponding position at unloaded static position. For each subject, the pattern of FO deformation was displayed, and the magnitude of maximum depression and reformation was extracted for each stance phase of walking. Finally, in order to determine significant differences between the deformation of sport and regular FO, statistical parametric mapping (SPM1D) was used to conduct non-parametric paired *t*-test on 2D deformation matrices. These matrices were generated from the deformation of points on FO plantar surface at each time frame for each subject.

## Results

The suggested technique to predict FO deformation was not only appropriate for simulating the loads and deformation during walking, but also accurate in terms of prediction error. As it was hypothesized, the AI model could also discriminate sport FO from regular FO.

### Validation

**Range of displacement.** The ranges of upward/downward displacement of each triad marker during walking session were a subset of its displacement during training session for all included subjects (Fig 2). For sport FO, the subjects generated an average range of [6.0±1.8; 6.1 ±1.8; 7.0±2.1 mm] on the medial front triad, [6.0±1.8; 6.3±1.9; 6.9±2.0 mm] on the medial middle, and [4.3±1.2; 4.8±1.6; 6.1±2.5 mm] on the medial back during walking compared to maximum amounts of [12.7; 14.6; 17.5 mm] on the medial front triad, [12.7; 13.7; 14.7 mm] on the medial middle, and [24.3; 23.6; 21.1 mm] on the medial back during training session. The back triad exhibited 12.1±3.7 mm during walking for its three markers versus 25 mm during training. The markers of lateral back triad displaced within a range of [7.2±2.1; 6.4±1.8; 5.6 ±1.7 mm] during walking, while they exhibited the maximum displacement of [18.8; 20.9; 23.9 mm] during training. In addition, the subjects generated an average range of [6.0±1.8; 4.4±1.3; 4.1±1.1 mm] displacement on three markers of lateral front triad during walking, while we imposed a maximum displacement of [12.6, 10.1, and 7.7 mm] on this triad during training session.

For regular FO, the ranges of displacement were lower for all triads during walking and training sessions compared to sport FO. On the medial side of FO, the subjects displaced the triad markers with the ranges of [5.3± 1.7; 5.3±1.8; 6.2± 2.3 mm] on the front, [5.3±1.7; 5.5 ±1.7; 6.2±1.8 mm] on the middle, and [2.7±0.7; 3.1±1.3; 4.0±1.5 mm] on the back during

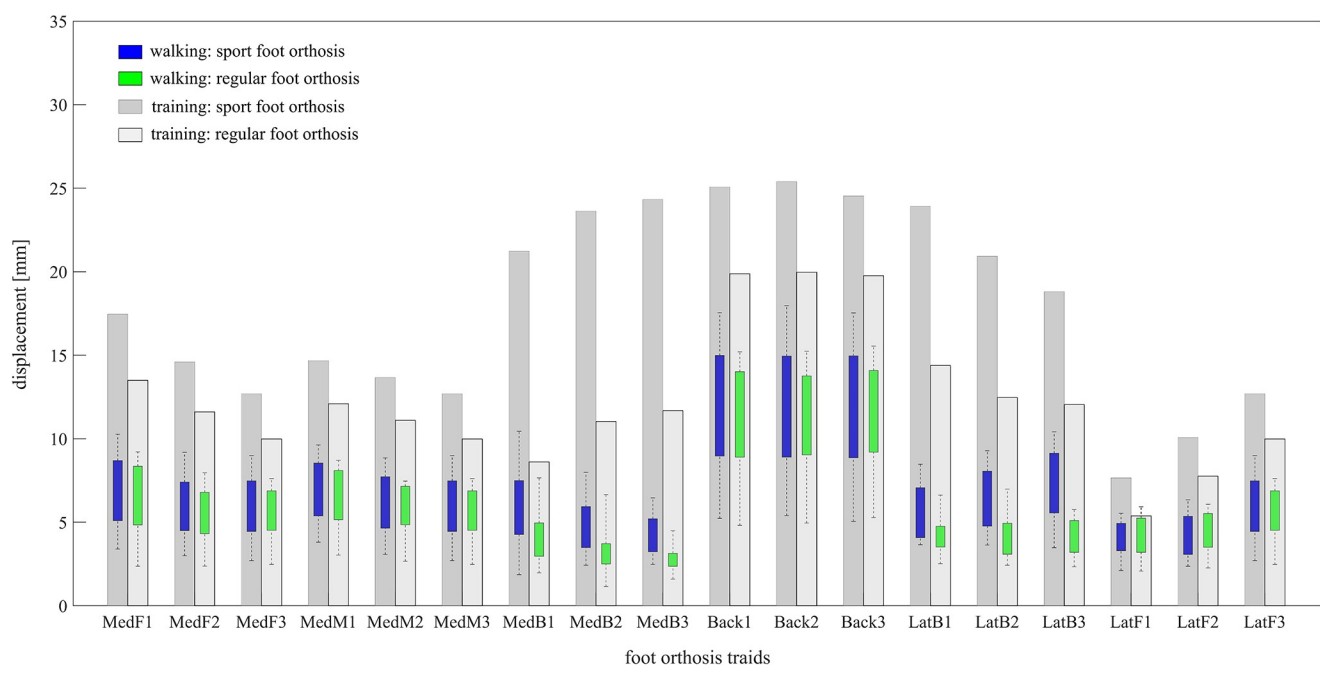

**Fig 2. Comparing the range of displacement for triad markers during training session and walking session for both sport and regular foot orthosis.**
The displacement for walking session is the range generated by all included subjects. The horizontal axis shows each of 3 markers on each triad namely MedF: Medial Front, MedM: Medial Middle, MedB: Medial Back, Back, LatB: Lateral Back, LatF: Lateral Front. The position on each triad marker can be observed in Fig 1.

walking. During training these triad markers were displaced with maximum ranges of [10.0; 11.6; 13.5 mm], [10.0; 11.1; 12.1 mm], and [11.7; 11.0; 8.6 mm], respectively. The back triad moved [10.7±3.5 mm] during walking compared to [19.9 mm] during training. The subjects showed an average of [4.0±1.1; 4.0±1.3; 4.2±1.1 mm] for the lateral back triad and [5.3±1.7; 4.3 ±1.3; 4.1±1.2 mm] for the lateral front triad during walking, while a maximum displacement of [12.0; 12.5; 14.4 mm] and [10.0; 7.8; 5.4 mm] was generated during training, respectively (Fig 2).

**Range of loading.** The results showed that the range of loading that was applied manually from stick to the FO during training session could cover the range of pressure that all subjects applied to each 10 regions of FO during walking for both FOs (Fig 3).

For sport FO, the subjects applied a range of 14.9±5.9 N for peak force (peak pressure multiplied by sensor area) on the medial front region, 23.7±5.6 N on the medial middle region and 25.7±6.8 N on the medial back region. During training, maximum loads of 53.7 N, 70.1 N, and 65.8 N were applied to the medial front, middle, and back regions, respectively. On the back region, the subjects applied a range of 21.7±12.2 N on medial back, 26.3±11.4 N on central back, and 19.9±10.7 N on the lateral back during walking compared to the exerted loads of 57.5 N, 53.5 N, and 62.0 N on these regions during training. On the lateral region, the range of peak force was 20.6±7.4 N, 25.3±6.2 N, 24.7±6.3 N during walking on the front, middle, and back regions compared to 63.3 N, 53.9 N and 64.1 N during training. For the middle region, maximum force of 26.1±7.9 N was applied during walking versus 73.2 N during training (Fig 3).

For regular FO, the average maximum forces for medial front, middle and back regions were 15.43±4.4 N, 28.2± 6.5 N and 28.2±11.3 N during walking versus 85.5 N, 104.4 N, and 102.2 N during training, respectively. On the back region, the subjects walked with applying maximum forces of 20.3±12.1 N on medial back, 21.9± 9.2 N on central back, and 23.4±13.5 N on lateral back, while the maximum stick load was 116.9 N, 52.0 N and 61.7 N on the

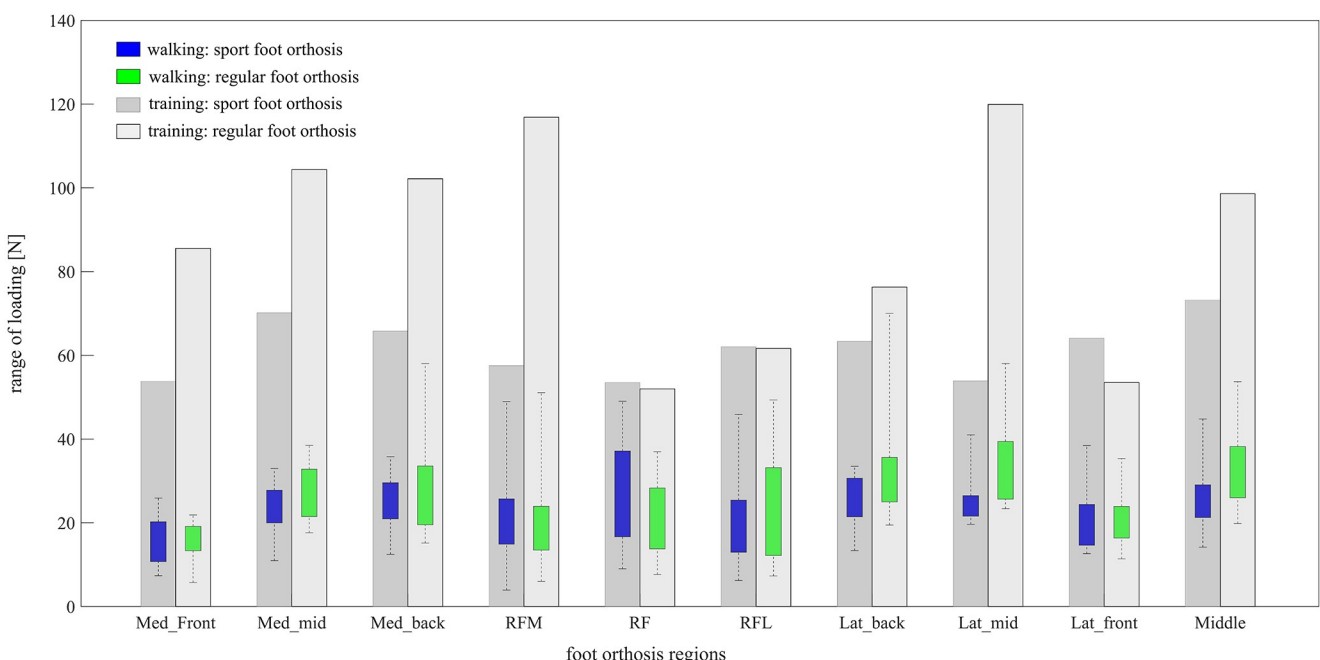

**Fig 3. Comparing the range of loading on different regions of foot orthosis during training session and walking session for both sport and regular foot orthosis.** The loading for walking session was calculated from the range of peak pressure generated by all included subjects.

corresponding regions. The FO experienced maximum forces of 21.6±7.1 N, 34.2±10.4 N, and 32.7±13.0 N during walking at lateral front, lateral middle, and lateral back versus 53.5 N, 119.9 N, and 76.3 N during training. The walking session showed maximum force of 32.3± 9.6 N on the middle region of FO, while a maximum force of 98.6 N was applied to this region during training (Fig 3).

**Prediction error.** The maximum deformation of the markers on plantar surface of FO ranged between 2.2 mm and 10.5 mm for sport FO and between 0.9 mm and 8.7 mm for regular FO with the largest deformation under the medial arch, lateral arch, and middle regions (Fig 4c). The mean and standard deviation RMSE for the 55 markers on FO plantar surface was 0.3± 0.1 mm (95% confidence interval for RMSE = [0.31, 0.36] mm) for sport FO, and 0.6 ± 0.1 mm (95% confidence interval for RMSE = [0.53, 0.61] mm) for regular FO. The reconstruction error was higher for regular FO compared to sport FO (Fig 4a). In addition, the

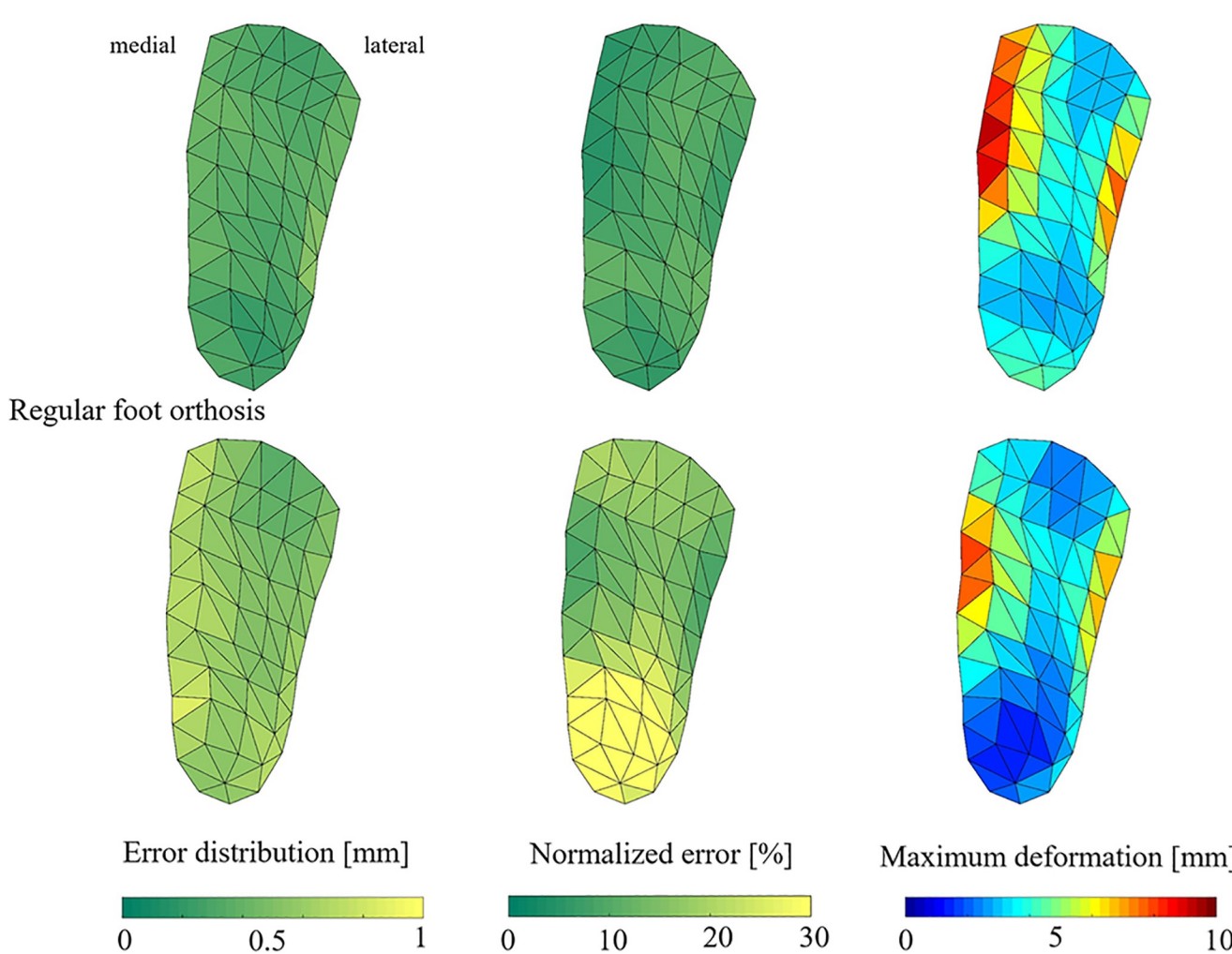

**Fig 4. Distribution of reconstruction error for sport and regular foot orthosis (a) Colormap of prediction error on plantar surface of foot orthosis [mm]; (b) Colormap of prediction error normalized to maximum deformation [%]; (c) maximum deformation on plantar surface of foot orthosis for test set [mm].** To show the error distribution on foot orthosis surface, the prediction error for each grid was calculated as the average error of its vertices.

normalized error was 8.2± 3.0% for sport FO and 20.6±12.6% for regular FO (Fig 4b). The highest NRMSE was observed on the back and medial/ lateral back regions of both FOs.

## Walking

Figs 5–7 showed that subjects generated different magnitude and patterns of deformation across the plantar surface of sport compared to regular FO during walking. Small range of variability was observed between subjects in each group of FO, mostly for the magnitude of deformation.

In general, the collapse of the foot medial arch imposed depression on the medial region of sport FO from heel strike to midstance (50% stance phase), where it started to reform until toe off. The lateral region of FO showed a reverse deformation compared to medial region. A reformation of FO under lateral arch from heel strike to either flatfoot or midstance was followed by depression until toe off (Fig 5). The middle region of FO exhibited depression from heel strike to toe off with shifting its maximal depression from medial to lateral side during stance phase. The median of maximum depression varied from -5.6 to -10.6 mm, and the maximum reformation from 0.2 to 3.7 mm between subjects for sport FO (Fig 6a).

For the regular FO, the depression of medial region from heel strike to either flatfoot or midstance was followed by a reformation until toe off with smaller range compared to sport FO (Fig 7). The lateral region of regular FO showed depression during either the whole stance phase or from heel strike to heel off. The depression on the lateral region was mostly focused on the frontal region rather than distributing in the whole lateral region in contrary to sport FO. The depression in the middle region of FO was mostly occurring from heel strike to heel off, which was accompanied with a shift from medial middle to lateral middle by advancing in stance phase. The median range of maximum depression changed from -4.0 to -6.8 mm between subjects, where they varied from 1.1 to 4.5 mm for maximum reformation (Fig 6b).

The average range of depression/reformation was [-7.7 to 0.5] mm for sport FO *versus* [-3.9 to 2.5] mm for regular FO (Fig 8). Statistical analysis showed significant differences in

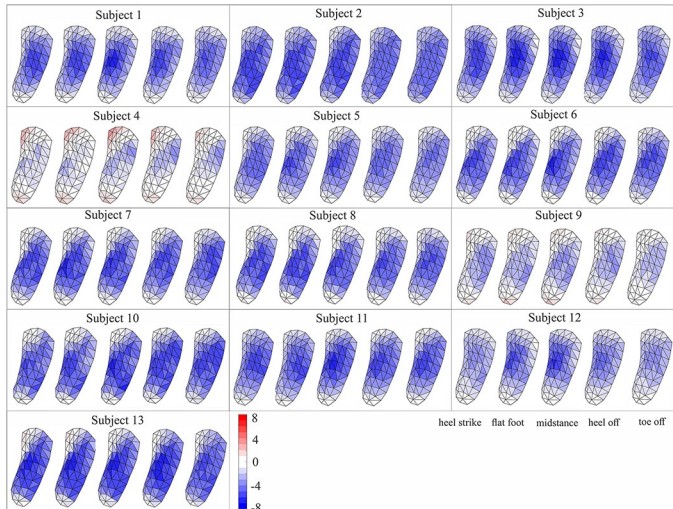

**Fig 5. Colormap of depression/reformation of sport foot orthosis during different key events of stance phase of walking for each subject.** The negative values show depression and positive values show reformation of FO. To show the deformation on FO surface, the deformation for each grid was estimated as the average deformation of its corresponding vertices.

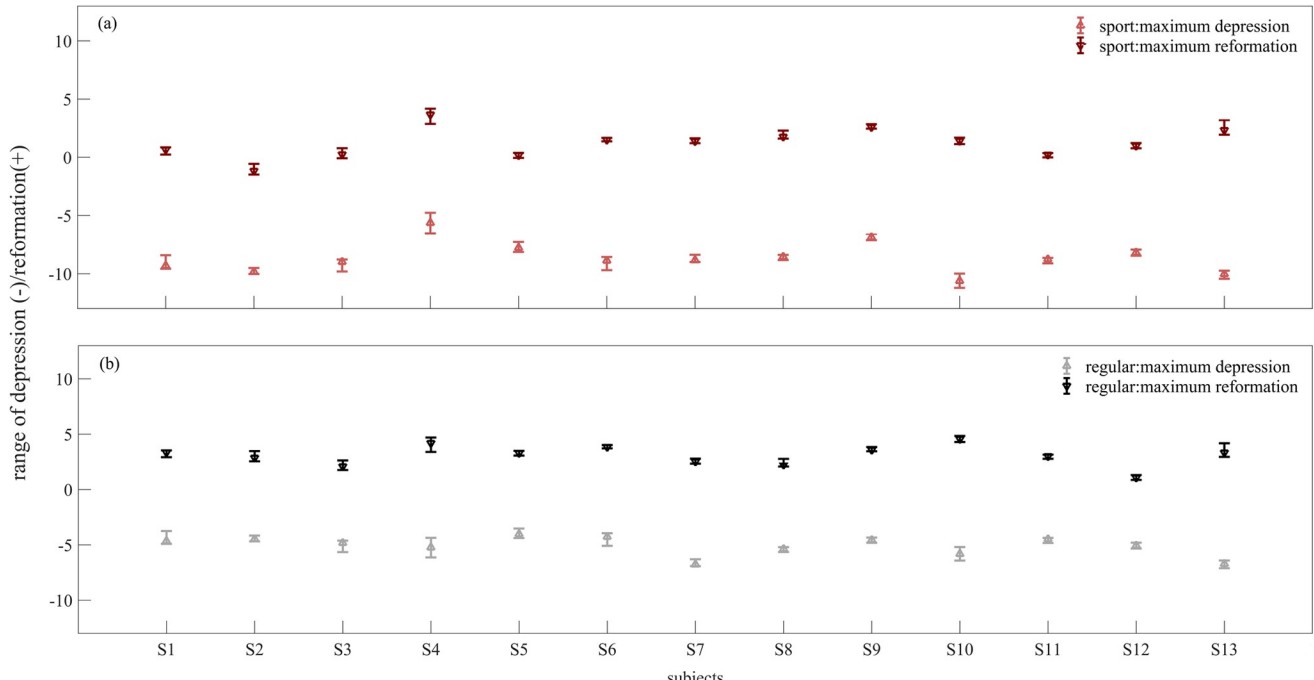

**Fig 6. The range of maximum depression and maximum reformation for each subject during stance phase of walking with (a) sport FO, (b) regular FO.**

deformation between sport and regular FO: the sport FO was more depressed on the middle region of FO, as well as the regions under medial and lateral arch compared to regular FO, from heel strike to toe off. In addition, the frontal extremity of lateral region and the back region showed more deformation in regular compared to sport FO.

## Discussion

Our study showed that FO deformation could be predicted from markers on removable triads fitted on the contour of the FO using an artificial neural network. The training session ensured a wide coverage of triads' displacement and applied loads on FOs that happens during walking, thanks to the designed setup. The average absolute prediction error on the plantar surface of FO was less than 0.6 mm for two FOs with different stiffness. In addition, the AI model was capable to distinguish the differences in the range and pattern of deformation between the sport and regular FO during walking for all included subjects. The model could also differentiate between subjects, as it could predict the small variability in the range of deformation as well as the gait moment of shifting depression onto reformation or vice versa.

The prediction error of markers on FO plantar surface during walking could not be calculated in similar way to the test set, since it was impossible to get makers' trajectories inside the shoes. Therefore, the output results of this study was significantly dependent on the input data. Indeed, the prerequisite of reaching accurate predictions is the existence of good overlap between the input data in training and test sessions [22, 26, 27]. This aspect was considered by developing a setup through which FO deformation was controlled based on the known forces. In fact, the stick as the tool of manual loading was equipped to a load cell for capturing the applied forces as well as retroreflective markers to retrieve the location of loading on FO. Moreover, the FO was placed on wooden plate covered with a layer of midsole property in

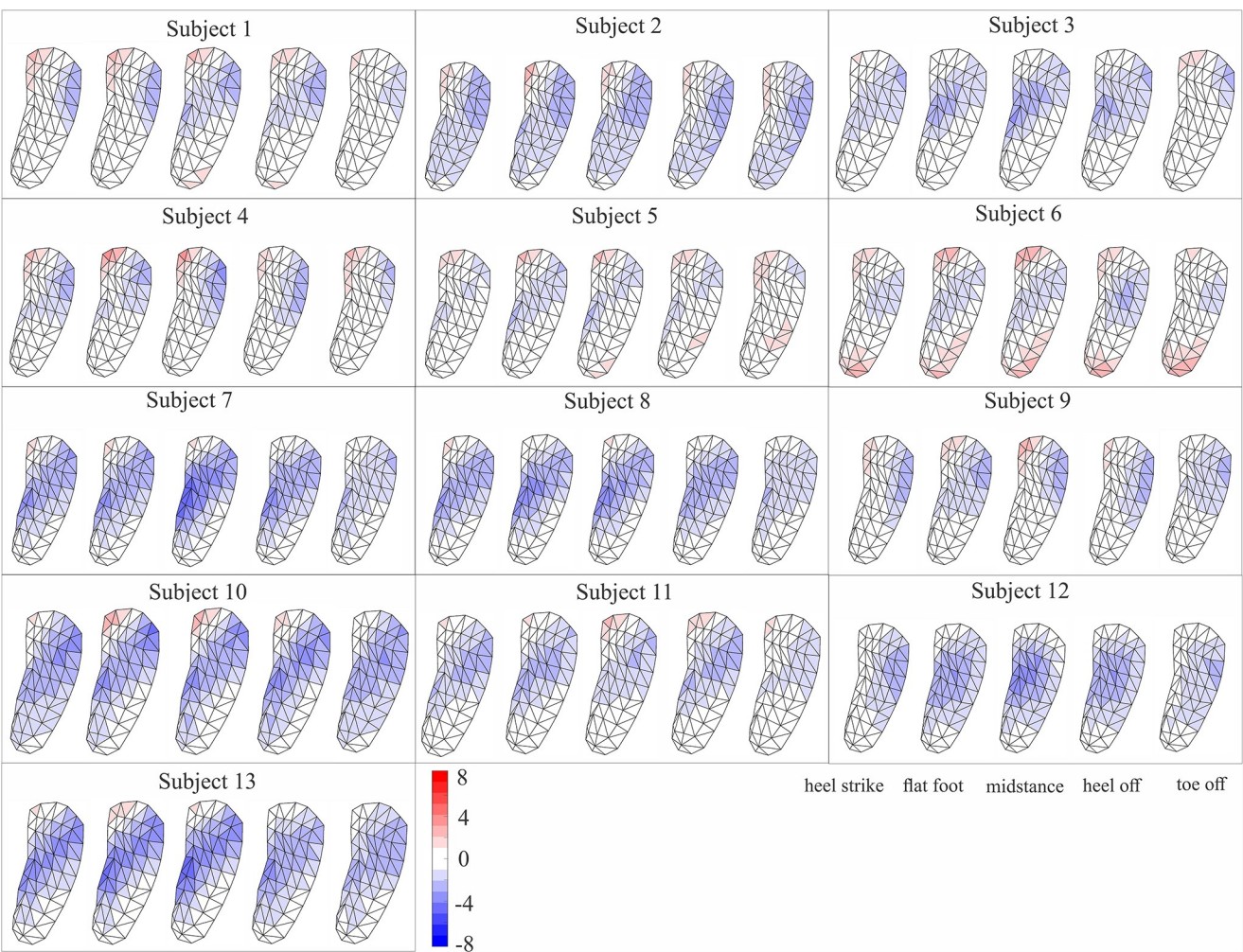

**Fig 7. Colormap of depression/reformation of regular foot orthosis during different key events of stance phase of walking for each subject.** The negative values show depression and positive values show reformation of FO. To show the deformation on FO surface, the deformation for each grid was estimated as the average deformation of its corresponding vertices.

order to simulate the support of FO inside the shoes. The heel part of FO was fixed on the plate to avoid sliding of FO due to load application. Heel cup is the supportive region of the FO to absorb the ground reaction force at foot strike and represents the most difficult region to be deformed under body weight loading [28–30]. Consequently, fixing heel cup could minimally affect the FO deformation. The results showed that the setup was robust in terms of producing the ranges of displacements for triad markers as well as the applied forces on several FO regions during training. In fact, the training data from this setup could cover all existing displacements and forces for both sport and regular FO during walking. Therefore, it can be assumed that the accuracy of FO deformation during walking would remain in the range of accuracy that was predicted for the test set data.

The absolute prediction error showed an average of 0.3 mm for sport and 0.6 mm for regular FO, which was distributed almost evenly on the whole surface of both FOs. This might indicate that our AI model has been capable to provide similar accuracy for predicting the position of markers on different regions of FO. The normalized error was maximum on the back and

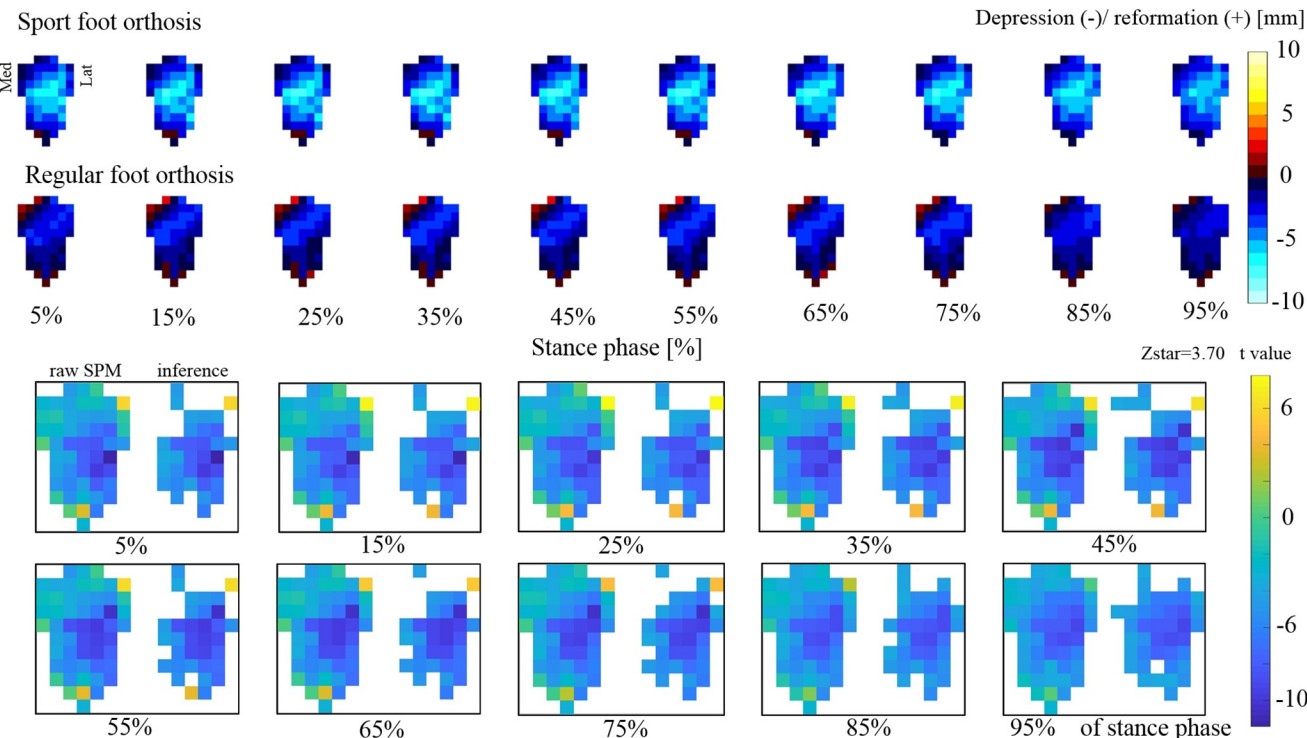

**Fig 8. For each 10 percent of walking (a) The average range and pattern of deformation for sport and regular FO during walking for all participants, (b) non-parametric paired t-test results using SPM1D to compare the deformation of sport versus regular foot orthosis.** Each subplot shows raw SPM (at left side) and inference (on right side).

medial/lateral back regions of regular FO. It could be due to the fact that the sport FO, and the regions under medial/lateral arch of regular FO were easier to deform due to the compliance and arch shape in contrast to medial/lateral back and back regions of regular FO during the training session. A previous research has reported 2.2 mm difference between arch deformation predicted from finite element model (9.9 mm) and experimental deformation (7.5 mm) in balanced standing position only [31]. Hence, in spite of maximum absolute error happening in the medial/lateral back regions of FO in this study, it was still lower than 1 mm during dynamic loading on FO, promising good prediction accuracy.

The deformation in several regions of FO during walking was extracted based on the predicted position of 55 markers spread out over the plantar surface of FO. The sport FO showed higher magnitude of deformation under medial arch region and middle FO region compared to regular FO, suggesting the capability of regular FO for providing higher level of support at medial arch. Furthermore, the reformation of sport FO under lateral region from heel strike to midstance was followed by a depression phase. In contrast, the regular FO exhibited depression under lateral arch and frontal arch regions during the whole stance phase or until heel off. Statistical results showed that regular FO exhibited significantly lower deformation during the whole stance phase, which can be inferred as providing more support by regular FO at medial/lateral arch and middle region of foot. For healthy normal feet, the lengthening of medial longitudinal arch lets the elastic structures of the arch to absorb and store energy at early stance. This energy would then be released during the late stance when the medial longitudinal arch recoils, and may provide enough power for propulsion [32]. In fact, the structure of FO works in series with the triangular architecture of medial longitudinal arch to ameliorate its stiffness

and facilitate the arch reforming [32, 33]. The extra support provided by regular FO might benefit flatfoot subjects by preventing their excessive collapse of medial arch during weight bearing phase [34]. Depression under medial arch was accompanied by depression on middle region as well as reformation under lateral arch region for sport FO. However, it was accompanied by depression of both middle and lateral regions for regular FO. Hence, the regular FO might have distributed the body weight on lateral regions more than sport FO. From mid-stance to toe off, both FOs were unloaded and reformed their shape under medial arch, while the FO deformation was transferred to the lateral middle and lateral arch regions. This could suggest that the stiffness of medial and lateral arches could control either foot posture such that the center of pressure is shifted medially/laterally, or foot function such that foot pressure distribution alters. It would subsequently change the forces and moment arm of the ground reaction force and consequently the ankle and knee joint moments. Excessive stiffness of FO can lead to further injuries in ankle, knee and hip such as osteoarthritis [35–37]. In order to optimize the functionality of FOs, it is necessary to customize the stiffness of FO based on foot motion, foot function and body weight. Reaching this goal primarily entails understanding which metrics for foot motion or foot function account for the most variability of FO deformation in different regions. The prediction of FO deformation will make it feasible for future studies to focus on this problem. The principal source of inter-subject variability might be due to using general prefabricated FOs for all included subjects, regardless of their weight and foot structure. It might also be due to different compensatory strategies that healthy subjects execute to deal with FO as the intervening tool between foot and shoes [38].

A limitation of this study was that point forces were applied during training session to see the markers displacements, and not a distributed force to simulate pressure as in walking. New setups could be developed to apply forces at different regions of FO plantar surface at the same time. Finite element analysis might be used as an alternative technique to generate the training session. However, each simulation with a single point load will take some hours, while it could not provide enough accuracy due to modeling simplifications. Another limitation was that relative orientation of triads were used rather than the position of triad markers in order to predict the position of markers on plantar surface of FO. It was found out that the triads were not fitted in the same depth of slots inside the FO contour during training session and walking session. Using the position of the markers as the input for AI model could subsequently propagate this error in the position of predicted markers. This experimental error was hardly possible to avoid due to the difficulty of measuring the depth of triads' insertions especially during walking session. However, this experimental error could be exempted by using the orientation of triads instead of the position of their markers. The relative orientation of each triad was calculated relative to back triad, because it had the minimal displacement. Finally, as foot orthosis is in direct contact with both foot and shoe sole, the deformation of FO is affected by the loading from foot and the boundary conditions imposed from shoe sole. It means that depending on the shape and mechanical properties of the shoe sole, we would change the degree of freedom for the range of deformation on the plantar surface of FO. In order to modulate the effect of shoe sole on the variability of FO deformation between subjects, standard shoes (New Balance 860 v8) were used for all subjects during walking. However, it is necessary to address the difference in the effect of shoe sole during training versus walking session as a limitation of this study. In the training session, the FO was fitted on a wooden plate covered with a soft material corresponding to a shoe midsole property. The movement of the shoe sole was therefore constrained in the training session. In contrast, the shoe sole during walking was capable to move and deform. This would lead to differences in the boundary condition applied from shoe sole on the deformation of FO. This limitation might be figured out in future studies by improving the setup design.

It is suggested that future studies look at the deformation of customized FOs on subjects with symptomatic feet. More between subject variations in the predicted deformation might be observed in such studies due to the different behaviors of customized FOs and their interaction with symptomatic feet. In addition, modifications in the design of slots and mechanical fit between triads and FO contour might be considered in future studies to reduce the experimental error in the position of triad markers. Finally, our findings were limited to FO deformation, while the correlation between FO deformation and plantar pressure as well as the correlation between FO deformation, foot kinematics, and arch flexibility is still unknown. Such results could bring advantages to mechanical and clinical aspects of customized FOs.

## Conclusions

Predicting FO deformation during dynamic activities is a novel and promising approach which reflects the direct interaction between foot and FO design. The results showed absolute error of less than one millimetre for predicting the deformation on plantar surface of both FOs. Our artificial intelligence model could discriminate between two FOs with different stiffness, i.e. "sport" versus "regular", by estimating different ranges and pattern of deformation during walking. Our model could also differentiate between different key events of stance phase. The trend of FO depression which shifted from medial arch to middle region and lateral arch by advancing in the stance phase seems realistic with biomechanical perspective. Inter-subject variability in FO deformation can be referred to different weight and foot shape. However, this variability was small due to the fact that our population had normal foot type, and wore the same prefabricated FOs. Further studies are needed to investigate how such information can be helpful to improve FO design for better functionality in terms of relieving pain and pathological symptoms.

## Supporting information

**S1 File. The pattern of foot orthosis depression/reformation for healthy subjects during walking with sport versus regular foot orthosis.**
(AVI)

**S2 File. Raw data for the training session of sport foot orthosis.** This Excel file consists three sheets in which the position of triad markers, the orientation of triad markers and the position of markers on plantar surface of foot orthosis are provided respectively.
(XLSX)

**S3 File. Raw data for walking with sport foot orthosis.** This Excel file consists two sheets in which the position of triad markers, and the orientation of triad markers are provided respectively for subject 1.
(XLSX)

**S4 File. The results of each participant during walking with sport foot orthosis.** This .mat file includes "DispEachPoint" and "DispEachPointMean" which shows the displacement of each predicted marker on foot orthosis plantar surface during stance phase of walking relative to its corresponding position in static non weight-bearing for each trial and the average of trials respectively. In addition, "loc_stance" and "loc_meanstance" show the location of each predicted marker during stance phase of walking. "peaks" and "peaksMean" represent the minimum (depression) and maximum (reformation) value of displacement during walking.
(MAT)

**S5 File. The results of each participant during walking with regular foot orthosis.** This .mat file includes "DispEachPoint" and "DispEachPointMean" which shows the displacement of each predicted marker on foot orthosis plantar surface during stance phase of walking relative to its corresponding position in static non weight-bearing for each trial and the average of trials respectively. In addition, "loc_stance" and "loc_meanstance" show the location of each predicted marker during stance phase of walking. "peaks" and "peaksMean" represent the minimum (depression) and maximum (reformation) value of displacement during walking.
(MAT)

**S1 Fig. The calibration session for extracting forces from load cell.**
(TIF)

**S2 Fig. The magnitude and location of loading applied from stick to deform foot orthosis in 10 regions: (a) sport foot orthosis, (b) regular foot orthosis.**
(TIF)

**S1 Table. The comfortable speed and step length of included participants.**
(DOCX)

## Author Contributions

**Conceptualization:** Maryam Hajizadeh, Benjamin Michaud, Mickaël Begon.

**Data curation:** Maryam Hajizadeh, Benjamin Michaud.

**Formal analysis:** Maryam Hajizadeh, Benjamin Michaud.

**Funding acquisition:** Mickaël Begon.

**Investigation:** Maryam Hajizadeh, Benjamin Michaud, Gauthier Desmyttere, Jean-Philippe Carmona.

**Methodology:** Maryam Hajizadeh, Benjamin Michaud, Gauthier Desmyttere, Jean-Philippe Carmona.

**Project administration:** Mickaël Begon.

**Resources:** Jean-Philippe Carmona, Mickaël Begon.

**Supervision:** Maryam Hajizadeh, Mickaël Begon.

**Validation:** Maryam Hajizadeh, Gauthier Desmyttere, Jean-Philippe Carmona.

**Visualization:** Maryam Hajizadeh, Benjamin Michaud, Gauthier Desmyttere.

**Writing – original draft:** Maryam Hajizadeh.

**Writing – review & editing:** Maryam Hajizadeh, Benjamin Michaud, Gauthier Desmyttere, Jean-Philippe Carmona, Mickaël Begon.

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
