## [Decision Letter · Decision Letter 0]

24 Mar 2020

PONE-D-20-03374

Predicting foot orthosis deformation based on its contour kinematics during walking

PLOS ONE

Dear Ms. Hajizadeh,

Thank you for submitting your manuscript to PLOS ONE. After careful consideration, we feel that it has merit but does not fully meet PLOS ONE’s publication criteria as it currently stands. Therefore, we invite you to submit a revised version of the manuscript that addresses the points raised during the review process.

We would appreciate receiving your revised manuscript by May 08 2020 11:59PM. To enhance the reproducibility of your results, we recommend that if applicable you deposit your laboratory protocols in protocols.io, where a protocol can be assigned its own identifier (DOI) such that it can be cited independently in the future. For instructions see: http://journals.plos.org/plosone/s/submission-guidelines#loc-laboratory-protocols

We look forward to receiving your revised manuscript.

Kind regards,

Arezoo Eshraghi, Ph.D.

Academic Editor

PLOS ONE

Journal Requirements:

2. Thank you for stating the following in the Competing Interests/Financial Disclosure* (delete as necessary) section:

We note that one or more of the authors are employed by a commercial company: Caboma, Montréal.

Reviewers' comments:

Reviewer's Responses to Questions

**Comments to the Author**

1. Is the manuscript technically sound, and do the data support the conclusions?

Reviewer #1: Yes

2. Has the statistical analysis been performed appropriately and rigorously? 

Reviewer #1: Yes

3. Have the authors made all data underlying the findings in their manuscript fully available?

Reviewer #1: Yes

4. Is the manuscript presented in an intelligible fashion and written in standard English?

Reviewer #1: Yes

5. Review Comments to the Author

Reviewer #1: A good paper, well written with a good hypothesis on a important subject.

My main concerns are following six:

1) Why did you select the comfortable speed in walking?

Walking speed influenced on the force for deformation of FO. Are walking speed and step length same in each conditions?

2) L 82．Why did you consider to be important for deformation of the FO?

3) L114．Why did you select the normal feet? Low arch feet have larger contact area of the FO than normal feet, and contact area influenced on the deformation of the FO. Moreover, persons using FO have the flat foot deformity.

4) What definitions as the normal feet? Asymptomatic? No deformation?

5) How did you consider the bending the sole( outsole) of the shoe during walking? I think you estimate the deformation of the FO, however your methods estimate the deformation of the FO and outsole.

6) Why did you select to compress using stick? When weight loaded on the FO, FO did not apply the compression force in “point”. The deformation of the FO is influenced by how to applying the force. Can we assume that the difference in the way applying force has no effect?

Nevertheless, this is a good contribute to the assessment of this "gray" clinical important of foot and ankle treatment.

6. PLOS authors have the option to publish the peer review history of their article (what does this mean?). If published, this will include your full peer review and any attached files.

Reviewer #1: No

---

## [Author Response · Author response to Decision Letter 0]

13 Apr 2020

Reviewers' comments:

Reviewer's Responses to Questions

Comments to the Author

1. Is the manuscript technically sound, and do the data support the conclusions?

Reviewer #1: Yes

Answer: Many thanks for your kind consideration and for the time you assigned for reviewing our manuscript.

2. Has the statistical analysis been performed appropriately and rigorously?

Reviewer #1: Yes

 3. Have the authors made all data underlying the findings in their manuscript fully available?

Reviewer #1: Yes

4. Is the manuscript presented in an intelligible fashion and written in standard English?

Reviewer #1: Yes

5. Review Comments to the Author

Reviewer #1: A good paper, well written with a good hypothesis on a important subject.

My main concerns are following six:

1) Why did you select the comfortable speed in walking?

Walking speed influenced on the force for deformation of FO. Are walking speed and step length same in each conditions?

Answer: Based on the protocol of this study, within-subject speed was the same, where between-subject speed could be different depending on each participant comfortable pace. The results of previous literature were considered to choose this option for designing our protocol. Previous research showed that at speeds either slower or faster than preferred speed the variability increases in young adults during treadmill walking [1]. In addition, it has been observed that each individual chooses a comfortable speed at which he reduces the energy consumption by parallelizing the passive mechanical properties of the legs with muscle activation. More stable condition has also been attributed to preferred walking speed [2-5]. On the other hand, some studies have addressed changes in gait pattern as an effect of walking speed [6]. However, as paired t-test is used in this study to look at the effect of foot orthosis rigidity, we believe that within-subject variability plays more important rule than between-subject differences for the purpose of this study. 

The following sentences were added to manuscript for more clarification:

“Each participant was asked to walk on a treadmill for 5 minutes for habituation, where his comfortable speed was acquired for the following measurements. Then, the participant walked for 3 minutes at this acquired speed for each sport and regular FO condition. A rest period of 5 minutes was given between conditions to avoid fatigue effects. The last 30 s of each trial was used for further processing.” [lines 161-168]

In addition, the speed of walking and step length for each subject was added to the supporting information (S1 Table) as following:

Subject Speed Step length [m]

#No. [m/s] Sport Regular

1 0.8 0.53 0.52

2 1.1 0.39 0.46

3 0.7 0.48 0.46

4 1.0 0.56 0.55

5 1.0 0.48 0.47

6 0.9 0.52 0.52

7 1.1 0.59 0.59

8 1.0 0.56 0.53

9 1.0 0.60 0.64

10 1.0 0.57 0.56

11 1.0 0.51 0.51

12 0.8 0.49 0.50

13 1.2 0.62 0.60

mean ± SD 1.0 ± 0.1 0.53 ± 0.06 0.53 ± 0.05

2) L 82．Why did you consider to be important for deformation of the FO?

Answer: We thank Reviewer #1 for asking this question. To our view, it is important to know FO deformation in order to improve the design of FO based on the biomechanical demands of different foot regions during dynamic activities. The primary requirement to achieve this goal was to validate the prediction of foot orthosis deformation on its whole plantar surface. FO deformation would then be used as a metric that represents the dynamic behavior of foot orthosis design. The next step would be to look at the interaction between foot biomechanics (kinematics and plantar pressure distribution) and the dynamic behavior of FO, i.e. FO deformation. Through this step, we could find out which metrics of foot biomechanics account for variation in FO deformation in each region of FO. Finally, this will help us to adapt the design of FO in each region based on the corresponding foot biomechanics and the level that we are interested to change based on the foot abnormality. 

The importance of estimating FO deformation has been better addressed in Introduction [Lines 63-68] and the future implication is pointed out in Discussion [Lines 574-578].

3) L114．Why did you select the normal feet? Low arch feet have larger contact area of the FO than normal feet, and contact area influenced on the deformation of the FO. Moreover, persons using FO have the flat foot deformity.

Answer: We appreciate Reviewer #1 for pointing out this important issue. In fact, the main aim of this manuscript was to (a) present and apply our artificial intelligence approach to predict the deformation of foot orthosis, and (b) to see if this technique is capable to differentiate between different rigidities of foot orthosis. It was assumed that healthy subjects with normal foot structure who walk with the same general 3D printed foot orthosis would impose less inter-subject variability compared to flatfoot subjects with abnormal foot posture (pronated foot and low medial arch). Therefore, healthy subjects were targeted for this study to more explicitly focus on the deformation of foot orthosis rather than incorporating the effect of abnormal foot deformity into the FO deformation. In the next step of this project, the validated technique would be implemented on subjects with flat foot deformity to find out their foot orthosis deformation during walking with their customized 3D printed foot orthosis.

4) What definitions as the normal feet? Asymptomatic? No deformation?

Answer: The inclusion criteria for participants was to be free from any limb injuries at the time of testing and having no known history of foot structural abnormalities or pathologies. The subjects were asked whether they have ever used foot orthosis or therapeutic insoles for any reason of pain or foot injuries especially flatfoot deformity. In addition, two observers, GD and MH, had to examine and include the subjects with normal medial arch during weight bearing/non weight bearing position and normal rearfoot orientation relative to tibia long axis.

These sentences were added to the Methods section [lines 116-124].

5) How did you consider the bending the sole (outsole) of the shoe during walking? I think you estimate the deformation of the FO, however your methods estimate the deformation of the FO and outsole.

Answer: This is a good question to be brought up. The triad markers were attached to FO contour, and they did not have any contact with the shoe sole. As a result, we believe that what is measured in this study is the deformation of the FO and not a combination of shoe sole and FO deformation. However, it is true that FO is in direct contact with both foot and shoe sole. Therefore, the deformation of FO is affected by the loading from foot as well as the boundary conditions imposed from shoe sole. It means that depending on the shape and mechanical properties of the shoe sole, we would change the boundary condition and the degree of freedom for the range of deformation on the plantar surface of FO. For example, a flexible shoe sole might provide more freedom for the vertical displacement of FO where a rigid FO might limit its displacement. In addition, a flat shoe sole provides different degree of freedom for the FO compared to a shoe sole with contour shape. The following sentences were added to the discussion to address this issue: 

“As foot orthosis is in direct contact with both foot and shoe sole, the deformation of FO is affected by the loading from foot and the boundary conditions imposed from shoe sole. It means that depending on the shape and mechanical properties of the shoe sole, we would change the degree of freedom for the range of deformation on the plantar surface of FO. In order to modulate the effect of shoe sole on the variability of FO deformation between subjects, standard shoes (New Balance 860 v8) were used for all subjects during walking. However, it is necessary to address the difference in the effect of shoe sole during training versus walking session as a limitation of this study. In the training session, the FO was fitted on a wooden plate covered with a soft material corresponding to a shoe midsole property. The movement of the shoe sole was therefore constrained in the training session. In contrast, the shoe sole during walking was capable to move and deform. This would lead to differences in the boundary condition applied from shoe sole on the deformation of FO. This limitation might be figured out in future studies by improving the setup design.” [lines 550-566]

6) Why did you select to compress using stick? When weight loaded on the FO, FO did not apply the compression force in “point”. The deformation of the FO is influenced by how to applying the force. Can we assume that the difference in the way applying force has no effect?

Answer: Thanks for pointing out to this limitation of our study, which is addressed in the discussion section [lines 531-538]. A 6-mm diameter stick was used in this study to apply point forces during the training session rather than the real distributed force that exist during walking. During training session, 55 retroreflective markers were taped on the small plantar surface of foot orthosis. The load application tool to deform the FO had to stimulate the loads during walking, in addition to not hiding the markers from the view of VICON cameras, which was the main challenge during data collection. For example, designing a wider stick with multiple tips (like a brushing hair) could stimulate the distributed load, in the cost of missing the position of a lot of markers. 

However, some points were considered in this study to make up for this limitation. During data collection, several repetitive loads were applied on different regions of FO using the stick which was equipped to a load cell for capturing the applied forces as well as retroreflective markers to retrieve the location of loading on FO (S2 Fig). This enabled us to check whether loading during the training session replicates what happens during walking. Although we cannot assume that the way of applying load has no effect, we showed that the range of loading that was applied manually from stick to the FO during training session could cover the range of pressure that all subjects applied to each 10 regions of FO during walking for both FOs (Fig 3). 

Nevertheless, this is a good contribute to the assessment of this "gray" clinical important of foot and ankle treatment.

6. PLOS authors have the option to publish the peer review history of their article (what does this mean?). If published, this will include your full peer review and any attached files.

Reviewer #1: No

References: 

1. Kang, H.G., J.B.J.G. Dingwell, and posture, Separating the effects of age and walking speed on gait variability. 2008. 27(4): p. 572-577.

2. Holt, K.G., J. Hamill, and R.O.J.H.M.S. Andres, The force-driven harmonic oscillator as a model for human locomotion. 1990. 9(1): p. 55-68.

3. Holt, K.G., et al., Predicting the minimal energy costs of human walking. 1991. 23(4): p. 491-498.

4. Jordan, K., et al., Walking speed influences on gait cycle variability. 2007. 26(1): p. 128-134.

5. Margaria, R., Biomechanics and energetics of muscular exercise. 1976: Oxford University Press, USA.

6. Fukuchi, C.A., R.K. Fukuchi, and M.J.S.r. Duarte, Effects of walking speed on gait biomechanics in healthy participants: a systematic review and meta-analysis. 2019. 8(1): p. 153.

---

## [Decision Letter · Decision Letter 1]

21 Apr 2020

Predicting foot orthosis deformation based on its contour kinematics during walking

PONE-D-20-03374R1

Dear Dr. Hajizadeh,

We are pleased to inform you that your manuscript has been judged scientifically suitable for publication and will be formally accepted for publication once it complies with all outstanding technical requirements.

With kind regards,

Arezoo Eshraghi, Ph.D.

Academic Editor

PLOS ONE

Additional Editor Comments (optional):

Reviewers' comments:

Reviewer's Responses to Questions

**Comments to the Author**

1. If the authors have adequately addressed your comments raised in a previous round of review and you feel that this manuscript is now acceptable for publication, you may indicate that here to bypass the “Comments to the Author” section, enter your conflict of interest statement in the “Confidential to Editor” section, and submit your "Accept" recommendation.

Reviewer #1: All comments have been addressed

2. Is the manuscript technically sound, and do the data support the conclusions?

Reviewer #1: Yes

3. Has the statistical analysis been performed appropriately and rigorously? 

Reviewer #1: Yes

4. Have the authors made all data underlying the findings in their manuscript fully available?

Reviewer #1: Yes

5. Is the manuscript presented in an intelligible fashion and written in standard English?

Reviewer #1: Yes

6. Review Comments to the Author

Reviewer #1: (No Response)

7. PLOS authors have the option to publish the peer review history of their article (what does this mean?). If published, this will include your full peer review and any attached files.

Reviewer #1: Yes: Shintarou Kudo

---

## [Editor Report · Acceptance letter]

24 Apr 2020

PONE-D-20-03374R1 

Predicting foot orthosis deformation based on its contour kinematics during walking 

Dear Dr. Hajizadeh:

I am pleased to inform you that your manuscript has been deemed suitable for publication in PLOS ONE. Congratulations! Your manuscript is now with our production department. 

With kind regards,

on behalf of

Dr. Arezoo Eshraghi 

Academic Editor

PLOS ONE